# Use of *Bacillus pretiosus* and *Pseudomonas agronomica* for the Synthesis of a Valorized Water Waste Treatment Plant Waste as a Biofertilizer Intended for *Quercus pyrenaica* L. Fertigation

**DOI:** 10.3390/biology14070902

**Published:** 2025-07-21

**Authors:** Diana Penalba-Iglesias, Marina Robas-Mora, Daniel González-Reguero, Vanesa M. Fernández-Pastrana, Agustín Probanza, Pedro A. Jiménez-Gómez

**Affiliations:** Department of Pharmaceutical Science and Health, CEU San Pablo University, Montepríncipe Campus, Ctra. Boadilla del Monte Km 5.300, 28668 Boadilla del Monte, Madrid, Spain; diana.penalbaiglesias@ceu.es (D.P.-I.); pedro.jimenezgomez@ceu.es (P.A.J.-G.)

**Keywords:** biofertilizers, valorization, WWTP, PGPB, *Quercus pyrenaica*, antibiotic resistance, cenoantibiogram, reforestation, degraded soil

## Abstract

Oak trees (*Quercus* spp.) are typically found in acidic, mature soils characteristic of the Mediterranean region, particularly within sub-Mediterranean and sub-Atlantic ecosystems. However, the conservation status of many oak groves is currently under serious threat and experiencing significant degradation. The application of plant growth-promoting bacteria (PGPB) strains, such as *Bacillus pretiosus* and *Pseudomonas agronomica*, has emerged as a promising strategy for the restoration of forest cover. These bacterial strains have previously demonstrated their effectiveness and viability within the rhizosphere during the entire fertigation period of host plants. Given the increasing scarcity of productive land, it is imperative to implement efficient strategies aimed at rehabilitating degraded natural areas. In this context, circular economy-based approaches, including the reutilization of organic chemical matrices derived from wastewater treatment plant (WWTP) by-products, have proven to be highly effective. The present study proposes a pot-based experimental approach to evaluate a biofertilizer formulated from the integration of WWTP-derived waste and the PGPB strains, *B. pretiosus* and *P. agronomica*. The biofertilizer exhibited a significant capacity to enhance the growth of *Quercus pyrenaica*, while simultaneously reducing the minimum inhibitory concentration (MIC) of the native soil microbia communities, without disrupting rhizospheric microbial diversity.

## 1. Introduction

Oak trees (*Quercus* spp.), like the K used in this study, are typically found in acidic, mature soils characteristic of the Mediterranean region, particularly within sub-Mediterranean and sub-Atlantic ecosystems, with abundant rainfall and humid summers [1,2]. The state of conservation of many *Q. pyrenaica* groves is seriously threatened and deteriorated, generating the syndrome of depletion due to regrowth. In addition, global warming poses additional challenges, as a reduction in the area occupied by sub-Mediterranean territories is expected, which will negatively affect the *Q. pyrenaica* groves [2,3]. A possible decline of the species is forecast, especially in areas of low altitude and latitude. The loss of biodiversity and forest cover, caused by pollution from industry, agriculture, and logging, favors this loss of diversity and biomass, crucial in the biogeochemical cycle of carbon and in the maintenance of environments with high ecological value [4]. To reverse this process of reduced diversity, the United Nations Strategic Plan for Forests 2017–2030 pursues the increase in forest area by 3% by 2030 [5]. Among the measures proposed for the restoration of these soils, the incorporation of organic matter to improve their structure and biological diversity stands out [6,7]. After a process of deforestation due to fire, logging, or pollution, there is an alteration of the organic matter of the soil, both in its physicochemical characteristics and in its capacity for decomposition [8]. Until now, most soil recovery approaches were committed to directly restoring vegetation cover, considering the state of soils by their chemical characteristics (pH, organic matter content, carbon content, and macronutrient content, among others) and their physical properties (structure, porosity, and nutritional deficits caused by fires). The biological properties of the soil have been neglected in many cases, despite showing an evident sensitivity to deforestation processes. Soil microbiota contributes positively to plant recovery through the restoration of key soil metabolic and ecosystem functions [9]. Therefore, recovery techniques must pay attention both to the restoration of vegetation and to the recovery of microbial communities in the soil and its physicochemical environment, helping to ensure a re-establishment of the complex relationships between plants and their underground environment [10]. This more holistic and integrative approach is expected to enhance the quality and reliability of ecosystem restoration data, surpassing the limitations of conventional methodologies [11]. Among the most promising strategies is the incorporation of organic amendments in the form of biofertilizers, which supply a complex nutritional matrix in addition to introducing beneficial microbial communities that improve nutrient cycling and promote plant growth [12,13,14]. These treatments have the potential to accelerate the restoration of the physicochemical and biological properties of the soil to levels similar to those of climatic phytosociological stages [15,16]. Organic fertilizers are nutrient sources of biological origin, obtained through the decomposition or transformation of organic materials. Valorization, based on the use of plant and animal waste or their by-products, makes it possible to enrich the soil and provide nutrients to plants. Therefore, these fertilizers of valorized origin offer a sustainable alternative for soil fertilization, as long as they do not contain pathogenic substances or microorganisms that contribute to the pollution of the environment. Organic fertilizers also have a positive impact on soil structure and biodiversity, as they promote aggregate formation and microbial activity, thus improving the soil’s ability to retain water and nutrients [17,18]. In particular, biofertilizers are formulated from a chemical matrix to which live microorganisms are added (mainly plant growth-promoting bacteria (PGPB) and fungi) that improve plant nutrition by mobilizing or increasing the availability of nutrients in soils [19]. These biofertilizers are one of the main tools to improve crop productivity and soil fertility, as the metabolism of the incorporated bacteria (mediated by hydrolytic enzymes, such as cellulases, proteases, and lipases) allows mineralization and increases the bioavailability of essential nutrients, such as nitrogen, phosphorus, and potassium that would otherwise be in complex forms not accessible to plants [20]. In addition, they improve soil structure and fertility by producing phytohormones that induce greater tolerance to abiotic stresses, including drought, salinity, and extreme temperatures, in plants [21]. The genera of PGPB most widely used for this purpose are *Bacillus* spp. and *Pseudomonas* spp. [22,23]. It is important that the strains used for the synthesis of biofertilizers for further use in the natural environment do not contain transmissible antibiotic resistance genes (ARGs) or virulence factor coding genes [24,25]. Thus, it is common for the use of a bacterium without antibiotic resistance mechanisms to have a positive effect on the soil microbial community that hosts it [26,27,28]. A useful tool to assess this impact is the cenoantibiogram [26]. Such analysis is essential to prevent or minimize the reserve and spread of resistances that may affect human, animal, and environmental health. This knowledge contributes to agricultural practices preventing the spread of antibiotic resistance mechanisms, minimizing the possibility of transferring these resistances to other soil microorganisms and the food chain. This fact reduces the potential risk to animal and human health by promoting a “One Health” vision of soil ecosystems [14,29,30].

Likewise, the observed effects of promoting plant growth, improving nutritional composition, and minimizing the response of soil communities to antibiotics must coincide with the survival of the bacterial strains added to biofertilizers in the rhizospheres of the plants that host them [31,32,33]. A useful tool for assessing such impact is amplicon sequencing-based metagenomics for *16S rRNA*. This analysis allows population and genetic diversity studies to be carried out. In ecological research, this approach is also known as environmental genomics, ecogenomics, or community genomics [34,35]. In this regard, biofertilizers originating from recovered biological waste to which PGPB *Bacillus pretiosus* (C1) and *Pseudomonas agronomica* (C2) have been added have previously been successfully used in *Lupinus albus* var. Orden Dorado [27], demonstrating its viability in the rhizosphere throughout the plant’s fertigation period. Understanding the effect of these strains on plants, soil microbial communities, and their interactions is essential for assessing the impact on biodiversity and the safety of biofertilizer application in the restoration of degraded soils, such as those associated with *Quercus pyrenaica* ecosystems. Considering the aforementioned background, the introduction of biofertilizers based on PGPB could offer a sustainable solution for the recovery of degraded soils. Therefore, the present trial aims to evaluate prospectively, and prior to its field trial, the efficacy of the biofertilizer derived from the combination of *B. pretiosus* and *P. agronomica* strains on *Q. pyrenaica* seedlings. In the same way, a study of the impact of biofertilizer and strains on rhizospheric communities of inoculated plants was carried out.

## 2. Materials and Methods

### 2.1. Bacterial Strains

The strains tested belong to the species *Bacillus pretiosus* (C1) and *Pseudomonas agronomica* (C2). The selection of these PGPBs was carried out by the research group (MICROAMB) “Environmental microbial biotechnology” of the Faculty of Pharmacy of the CEU San Pablo University. These bacteria have a possible biotechnological interest due to plant growth-promoting characteristics [27]. Table 1 shows the main properties of the two used strains. Whole genome sequencing rules out the presence of virulence factors or transmissible antibiotic resistance genes.

### 2.2. Experimental Design: Irrigation Matrices (Chemical Treatment) and Biological Treatment (PGPB Strains)

For the growth test under laboratory conditions, 10 cm × 8 cm seedbeds with 35 cm × 18 cm high alveoli were used, placed in forest trays or leachate trays. The acorns were added and subsequently covered with a commercial peat-based substrate. The biological tests were carried out on seedlings of 1 sap of *Quercus pyrenaica*, supplied by IMIDRA (Madrid Institute of Rural, Agrarian and Food Research and Development). These seedlings came from seeds of native plants in the Community of Madrid, whose purpose is to repopulate forests to avoid genetic contamination. Each treatment was carried out on separate trays in order to avoid cross-contamination. The following three irrigation matrices (chemical treatment) were tested: water (W), organic waste from a WWTP, and the same sterilized waste (EDAR_ST). Each of them was tested with the following three biological treatments: control without inoculum (C0), supplemented with *Bacillus pretiosus* (C1), and supplemented with *Pseudomonas agronomica* (C2). Each trial was conducted in triplicate (n = 3). Each replicate was constituted with 20 replicates (seedlings) to guarantee sufficient biomass for further analysis.

### 2.3. Preparation of Bacterial Suspensions and Biofertilizers

#### 2.3.1. Preparation of Bacterial Suspensions

Starting from pure cultures of the C1 (*Bacillus pretiosus*) and C2 (*Pseudomonas agronomica*) strains in nutritional agar from Condalab^®^ (Madrid, Spain), a bacterial culture was prepared in LB liquid medium. After 48 h of growth, the bacterial density was checked at 0.5 on the McFarland scale (10^8^ u.f.c. mL^−1^) using UricultTM submersible paddles (Liofilchem srl, Roseto degli Abruzzi, Italy). This process ensured the standardization of the bacterial inoculum for its subsequent application in the final volume of the irrigation matrix.

#### 2.3.2. Preparation of Biofertilizers and Addition of PGPBs to WWTP Fertilizer: Fertilizer Formation

The WWTP (EDAR) waste (Industrias Cárnicas Villar, S.A., Soria, Spain) was used at a concentration of 1/512 (V_WWTP_/V_H2O_). The physicochemical composition of the organic waste can be seen in Table 2. For the preparation of the sterilized organic waste EDAR_ST, in order to eliminate microbiota, it was sterilized by autoclaving (121 °C, 20 min, 1 atm). The biofertilizer was prepared weekly (3 L, storage at 4 °C) to prevent bacterial growth and transformations. To ensure a microbial density of 0.5 McFarland in an irrigation matrix. For the synthesis of the biofertilizer, 100 mL of a 0.5 bacterial McFarland suspension of each strain, *Bacillus pretiosus* (C1) and *Pseudomonas agronomica* (C2), was added, except for the irrigation matrix, which only presented H_2_O. A total of 900 mL of H_2_O from the tap. Next, 20 mL of the organic waste was added.

#### 2.3.3. Plant Growth Conditions and Irrigation Regime

A 36-week trial was conducted under controlled laboratory conditions. The growth phase was carried out in a controlled-environment phytotron under a regulated photoperiod of 11 h light and 13 h darkness, a light intensity of 505 μmoles.m^−1^.s^−1^ (white and yellow light), a temperature of 25 °C ± 3 °C, and a relative humidity of 30% ± 5%. The irrigation regime was designed to simulate field conditions. Each alveolus was irrigated with an experimental volume of 85 mL (aerial irrigation) plus 500 mL to each leachate tray (capillary irrigation). Two waterings were carried out weekly for the duration of the experiment (from 4 January 2023 until collection on 2 October 2023), as follows: on Mondays with chemical and biological treatment and on Wednesdays with water, in order to maintain humidity.

#### 2.3.4. Harvesting, Total Biomass Measurement, and Nutritional Analysis

Once the 36 weeks of the trial were over, the harvest was conducted. This process involved the destructive extraction of the aerial and radical part of each plant. From the root fraction, a sample of rhizospheric soil (2g per seedling) was obtained for subsequent analysis. To determine total biomass, the harvested plants were left to dry at room temperature (22 °C ± 2 °C) for one week, recording the dry weight in grams (g). For nutritional analysis, the leaves were separated from the stem manually. Each fraction was divided into three replicates, each consisting of leaves or stems pooled from several individual plants. The samples were packaged and kept refrigerated for shipment to the Rock River Labs Spain analysis laboratory, located in Lalín, Pontevedra, Spain. Nutritional analysis was performed within 24 h of harvest. The parameters that were measured in the nutritional analysis were proteins (%DM), total amino acids (%CP), minerals (%DM), carbohydrate digestibility (%DM), sugars (%DM), fiber digestibility (%DM), and fatty acids (% Total FA).

### 2.4. Extraction of Soil Microbial Communities

For the extraction of the rhizospheric communities, the procedure described in Velasco et al. [36] was followed. To this end, 2 g of rhizospheric soil were suspended in 20 mL of sterile saline solution (0.45% NaCl) and homogenized using an Omni-Mixer homogenizer at 16,000 rpm for 2 min. It was then centrifuged at 690× *g* for 5 min with a Hettich Zentrifugen centrifuge model Mikro 22R (Hettich, Tuttlingen, Germany).

### 2.5. Cenoantibiogram

The soil extract obtained in saline solution (NaCl 0.45%) showed a density of viable microorganisms greater than 10^8^ u.f.c. mL^−1^, with an optical density (OD) of 0.5 on the McFarland scale. Subsequently, seeding was performed on Mueller–Hinton agar (Condalab^®^, Madrid, Spain) and the Minimum Inhibitory Concentration (MIC) was evaluated using antibiotic strips of ε test for several antibiotics, including amoxicillin (AML), amoxicillin-clavulanic acid (AUG), piperacillin (PP), piperacillin-tazobactam (TZP), imipenem (IMI), imipenem-EDTA (IMD), ciprofloxacin (CIP), nalidixic acid (NA), trimethoprim-sulfomethoxazole (TS), cefotaxime (CTX), and cefpirome (CR) (BioMérieux^®^, Marcy l’Etoile, France). The plates were incubated according to the manufacturer’s instructions, at 25 °C, and the minimum inhibitory concentration was quantified using the most restrictive inhibition halo as a reference.

### 2.6. Study of the Functional Diversity of the Microbial Community

From the soil extract, Biolog Eco^®^ plates were inoculated with 135 μL per well (31 wells with 30 different carbon sources and one control, in triplicate). The plates were incubated for 144 h, at 25 °C ± 2 °C, and their absorbance was measured at 595 nm, every 24 h using the Asys UVM340 plate reading equipment and the Micro WinTM V3.5 Software. With the results of the absorbance measurements, each value was corrected by subtracting the target (corrected absorbance). Next, the mean corrected absorbance of each replicate was calculated as the average of the 31 wells in the Biolog Eco^®^ plate (Biolog, Inc., Hayward, CA, USA). The value of AWCD (average well color development) [37] was plotted against the incubation time to obtain the growth curves of the microbial community in the wells of the plate. In these curves, the incubation time at which the growth of the microorganisms was starting the stationary phase was chosen, for subsequent multivariate analyses. Additionally, with the corrected absorbance values of the incubation moment chosen as AWDC, the metabolic diversity of each sample was calculated using the Shannon–Weaver diversity index.(1)$$H\left(m\right)=-\sum q_{i\;}\log_{2}\;q_{i}$$ where *q_i_* = n/N; where n is the corrected absorbance (AWCD) of each well, and N is the total absorbance of all wells.

### 2.7. Statistical Analysis

The study included an analysis of biometrics and plant nutrition based on different types of chemical and biological treatment. A one-factor ANOVA test was used to assess the variation in biomass dry weight in relation to different types of irrigation. Subsequently, a post hoc Duncan analysis was carried out to compare the means and determine the statistically significant variations in plant growth. Regarding nutritional parameters, a first ANOVA was performed to identify differences in the means depending on the type of irrigation. Only those parameters with statistically significant differences were subjected to a post hoc Duncan analysis to determine which treatment explained these deviations. In addition, a principal component analysis (PCA) was used as an exploratory technique to identify trends between the variables analyzed according to biological and chemical treatments. All these analyses were carried out using the SPSS v.29.0 program (IBM Corp, Armonk, NY, USA), which allowed a detailed evaluation of the relationships between the different factors studied and their effects on plant biometrics and nutrition in the context of the treatments applied.

### 2.8. Metagenomics

The composition and structure of the sampled microbial communities was evaluated by amplification and sequencing of the variable regions V3-V4 of the *16S rRNA* gene. Amplification was performed after 25 cycles of PCR. In this procedure, positive (CM) and negative (NC) controls were used to ensure quality control. The positive control is a mock community and was processed in the same way as the samples. The libraries obtained were sequenced using Illumina Miseq (300 × 2).

#### Bioinformatics Processing and Statistical Analysis

A schematic view of the process can be found in Figure 1. Demultiplexed crude readings, both forward and reverse, were processed using QIIME2 [38].

Beta diversity distance matrices were used to represent the PCA and to make ordination plots using the R software package version 4.2.0. The significance of the groups was tested using the Permanova and ANOSIM tests. The Permdisp test was used to identify location vs. scattering effects [39]. The significance threshold (*p*-value) was set at 0.05.

The taxonomic assignment of the phylotypes was performed using a Bayesian classifier [40] trained with the Silva version 138 database (full-length sequences at 99% OTUs) [41]. The differential abundance of taxa was tested using generalized linear models with negative binomial distribution. A generalized linear model was calculated using the R MASS v.7.3-54 package [42]. The significance threshold (*p*-value) was set at 0.05. The packages BiodiversityR v2.14-1, PMCMRplus v1.9.4, RVAideMemoire v0.9-8, and vegan v2.5-6 were used for the different conducted statistical analyses.

## 3. Results

To aid the interpretation and analysis of the key results, graphs are only presented for those variables that, after the application of the ANOVA test, showed statistically significant variations (*p*-value < 0.05), depending on the type of treatment (irrigation matrix or chemical treatment and biological treatment).

### 3.1. Biometrics

Figure 2 shows the results obtained after a biometric study of the stem, including (2A) weight (g) and (2B) length (cm).

Duncan’s post hoc analysis shows that there is a statistically significant improvement in stem weight, with the incorporation of treatment with the PGPB C2 strain in EDAR_ST matrix and with the C1 strain in a WWTP matrix, significantly increasing the stem weight, differentiating itself from the rest of the treatment and controls (W). In relation to stem length (Figure 2B), a statistically differential improvement has been found with the incorporation of strains C1 and C2 in a WWTP matrix and EDAR_ST for the PGBP C1 strain and in the EDAR_ST matrix for the C2 strain. These results suggest that the presence of bacteria in combination with WWTP has a significant effect on both the stem weight and length, with strain C1 in a WWTP matrix, as well as with a strain C2 with a matrix EDAR_ST.

### 3.2. Nutritional Analysis

A post hoc analysis was carried out on those variables that, after performing the ANOVA, showed statistically significant variations (*p*-value < 0.05), depending on the type of chemical and/or biological treatment applied.

Figure 3 shows the variables of the protein group that varied statistically according to the chemical and/or biological treatment, as follows: crude protein (%), soluble protein (%), and total amino acids (% dry mass, DM) (*p*-value < 0.05).

In terms of intertreatment in Figure 3A, there is a statistically differential improvement in crude protein content, regardless of the incorporation of PGPB strains, in all cases, when a matrix other than water (EDAR and EDAR_ST). In all cases, the incorporation of PGPB strains C1 and C2 improves crude protein content, except for treatment without PGPB strain in a matrix of the EDAR_ST. In relation to the content of soluble proteins, treatment with the EDAR_ST matrix and with the PGPB C1 strain significantly increases the amount of soluble protein, clearly differentiating it from all other treatments (EDAR) and controls (W). In terms of intertreatment in Figure 3B, there is no statistically differential improvement in crude protein content, regardless of the incorporation of PGPB strains, in all cases, when a matrix other than water (EDAR and EDAR_ST). In all cases, the incorporation of the PGPB strains C1 and C2 improves the soluble protein content, compared to the control without inoculum. This increase is significant in the case of irrigation with WWTP sterilized residue with the PGPB C1 strain. In any case, there are no significant differences within treatment, in irrigation with one strain or another. With respect to soluble protein content, there is no differential statistical improvement regardless of the incorporation of PGPB strains, in all cases, when a matrix other than water (EDAR and EDAR_ST). However, there is a difference with the treatment with the PGPB C1 strain and with the irrigation matrix of the EDAR_ST.

Figure 4 shows the amino acids that have shown statistically significant variations between the different treatments (*p*-value < 0.05).

Consistent with the observations in Figure 3, the incorporation of the chemical matrix into the irrigation treatments (EDAR and EDAR_ST) resulted in a significant increase in amino acid content, with a particularly notable enhancement in histidine levels (Figure 4). Similarly, intratreatment comparisons reveal an increase associated with the addition of either C1 or C2, relative to their respective non-inoculated controls. Notably, the treatments EDAR_STC1 and EDAR combined with both bacterial strains (C1 and C2) showed the most pronounced increases in total amino acid content relative to the other treatments.

PCAs of the nutritional variables in leaves (Figure 5) and stems (Figure 6), along with their corresponding loading factors, show how these variables are distributed along the *x*-axis and similarly grouped according to the applied biological treatments. In Figure 5 and Figure 6, only components 1 and 2 were represented, which together explain 75.55% of the variance in leaf and 66.87% in stem. In both figures of factors, it is observed that most of the variables studied are displaced towards the positive *x*-axis.

Therefore, in the comparison of PCA with their respective load factors, we found that individuals irrigated with C0 (in the red ellipse) have lower values compared to those inoculated with C1 and C2 in most of the variables studied. Thus, protein or carbohydrate content values stand out, which is also observable in Figure 3 and Figure 4.

The metabolic diversity of each sample was calculated using the Shannon–Weaver diversity index, and the differences were analyzed at 144 h. In most natural ecosystems, it typically ranges from 0.5 to 5. Values between 2 and 3 are considered normal, with values below 2 indicating low species diversity and those above 3 indicating high diversity.

Table 3 shows the results obtained from the diversity analysis using the Shannon statistic. As evidenced by the analysis of rhizospheric metabolic diversity, all treatments exhibit consistently high diversity levels, with an average Shannon index (H′) of 4.68, irrespective of the specific intervention applied. No significant differences were found between the treatments.

Figure 7 shows the grouping trends of the soil communities subjected to the different fertigation treatments, according to their phenotypic profile of resistance to the different antibiotics tested.

As shown in Figure 7A, there is a tendency for treated soils to be grouped according to biological treatment. Treatments inoculated with C1 and C2 strains cluster on the left side of the *x*-axis; although, they remain distinct from one another, whereas treatments lacking bacterial supplementation are grouped on the right. Comparing this distribution with the load factors in Figure 7B, it can be intuited that, due to the grouping of all antibiotics towards positive values of the *x*-axis, they capture the segregation of treatments lacking inoculum.

Table 4 and Table 5 represent a bioinformatic analysis in relation to the quality of the sampling. After the first sampling, 10,910 phylotypes were detected, demonstrating a great diversity. Moreover, singlets and doublets were eliminated to avoid noise in the data, because these are low-frequency phylotypes. To avoid bias introduced by unequal sequencing depths, all samples were subsampled to the same number of reads prior to comparison. These results indicate uniform and sufficient coverage between samples, with a medium frequency that ensures adequate depth for further analysis. The values of the first and third quartile reflect a consistent distribution of data, while the median and mean suggest that there are no significant biases in the coverage of the sequences. The quality of the data confirms that the correct bioinformatic analysis is appropriate.

Figure 8 presents the taxonomic diversity results using the unweighted Unifrac metric. This statistic only considers the presence or absence of a given OTU in the analyzed community.

Figure 8 shows three well-differentiated groups according to biological treatment. In the groups treated with the C1 and C2 strains, the point clouds appear very compact, indicating that they have a similar trend in terms of taxonomic division.

#### Relative Abundances at the Gender Level

Figure 9 and Figure 10 display differences in community composition irrespective of relative abundance, as visualized using bar plots.

In Figure 9, we find the abundance of taxonomic composition, highlighting the taxa of interest, C1 and C2. To simplify the figure, a simplification has been made.

Figure 10 shows how those treatments with C1 and C2 have a higher relative abundance of the genera *Bacillus* and *Pseudomonas* compared to those treatments without inoculum (C0).

## 4. Discussion

The processes of deforestation and soil degradation, as well as climatic alterations, increase the scientific–technical interest in developing new strategies that allow the adaptation of silvicultural species based on plant–microorganism symbiosis [43,44]. These strategies and tools aim to reduce the recovery times of degraded soils, based on the phenology and growth rates of plant species, in order to improve the efficiency of these processes [43,44]. The valorization of resources and the use of PGPBs adapted to each soil and host can contribute to improving the different qualities of the plant [45,46]. Among these PGPBs, the genera *Pseudomonas* spp. and *Bacillus* spp. [47,48,49] stand out. Previous trials using strains of *Bacillus firmus* and *Bacillus subtilis* show a dual function as plant growth promoters and biological control agents [49,50]. Likewise, the genus *Pseudomonas* associated with the root of different plant species plays an important role in promoting growth and protecting crops against plant pathogens and invertebrate pests [46]. These capacities are of interest for the use of both bacterial genera in the biotechnological industry as constituent components in the synthesis of biofertilizers. Similarly, regardless of the chemical matrix used in fertigation, the incorporation of the bacterial strains C1 (*Bacillus pretiosus*) and C2 (*Pseudomonas agronomica*) led to an increase in biomass production and stem length compared to their uninoculated control counterparts (C0). Furthermore, significant differences across all variables were observed when comparing plants treated with the C1 (*Bacillus pretiosus*) and C2 (*Pseudomonas agronomica*) strains in the WWTP and EDAR_ST matrices to their respective non-fertilized controls. In line with these observations, Puri et al. [51] observed that the use of certain PGPB strains of the genus *Bacillus* improves the growth, biomass production, and root elongation of pine plants relative to those without inoculum.

Moreover, WWTP waste used as a chemical matrix for irrigation contains various complex nutrients that PGPBs can biotransform into more bioavailable inorganic forms for plant uptake [52]. Therefore, when comparing the results of the plants subjected to the different chemical treatments, a greater impact on stem length in those that have used the sterilized matrix of EDAR_ST is observed. This may be explained by the elimination of potential phytopathogenic microorganisms from the residue, which could enhance the efficiency of resource biotransformation by the inoculated strains. In particular, the results obtained evidence of how the C1 strain (*Bacillus pretiosus*) is capable of stimulating plant growth both in EDAR and in EDAR_ST, having greater cost–benefit potential in its subsequent use as a biofertilizer. A similar effect was reported by Tang et al. [53], who worked with crop plants using WWTP water as a matrix. In that study, the water underwent alkaline thermal hydrolysis prior to application in order to enhance its efficacy as a fertilizer. These results postulate this fertilizer valorized for subsequent biotechnological tests in the field aimed at favoring the primo-colonization of degraded areas that allow favoring the development of more complex phytosociological stages. This is due to the enrichment of both the amount of organic matter and the number of soil microorganisms, factors that allow a greater fixation of essential nutrients for plant growth and, therefore, an increase in soil fertility [54].

On the other hand, the nutritional status of plants can serve as an indirect indicator of their adaptive capacity and resilience [54,55,56]. In the present study, it has been found that the incorporation of PGPB significantly improves many of the variables that define the nutritional quality of the plant after harvest. The differences in these nutritional variables increase their significance when the biofertilizer is used based on the raw WWTP organic waste added with either of the two tested PGPBs. The mineralizing microbial activity contributes to improving the bioavailability of the nutrients contained in the organic fertilizer by increasing their concentration in the soils [57,58]. Thus, in a general way we observe that the incorporation of both the C1 strain (*Bacillus pretiosus*) and the C2 strain (*Pseudomonas agronomica*) significantly increases the concentration of proteins and the digestibility of carbohydrates, minerals, and fatty acids in the composition of plants.

Beyond the significant impact that the incorporation of PGPB strains has on the nutritional and biometric status of the plant, it is important to study the effect that these exogenous bacteria may have on rhizosphere communities [59]. It is known that the incorporation of a microorganism can reduce the biological diversity of an ecosystem [60]. However, there are some exceptions to this postulate, given that in the field of microbial ecology, the introduction of a species into a system can lead to an increase in this diversity as long as there is no displacement of the rest of the taxa in the community [58]. The most frequently evaluated aspects of diversity are the richness measured in number of species, the proportional distribution of the number of individuals of each species, and the metabolic and functional diversity of the system [61,62,63,64,65]. These measurements are a way of describing ecological communities, in terms of dominance or equity, as another component of diversity [60,61,62].

In the field of microbial ecology, many indices have been proposed for measuring biodiversity. One of the most widely applied indices, also employed in the present study, is the Shannon index [66]. The results of the present study showed that the addition of biofertilizers does not induce a significant alteration of functional diversity. This may be due to the shielding effect of the soil, which is very rich in nutrients. Therefore, the stability of microbial diversity should be interpreted as an expected biological behavior [59]. Rhizospheric communities seem to host exogenous strains that promote different positive effects on the plant, without altering the functionality of the community [67].

On the other hand, the addition of a strain to a biological system can alter the antibiotic resistance profile of the soil community that hosts them [25,27]. Once incorporated into the soil, they can displace more resistant phenotypes, thereby reducing the overall MIC of the soil microbiota, alongside the horizontal transfer of these resistance traits [22,65]. Previous studies characterized the phenotypes of the PGPB strains used in this work and reported low minimum inhibitory concentrations (MICs) when exposed to several antibiotics commonly used in human and veterinary medicine [27]. Furthermore, in the present work, it was observed how the addition of PGPB strains induces a decrease in the MIC of the soil community. Its analysis from a “One Health” perspective allows the interpretation of a bioprotective effect by minimizing the horizontal transmission of gene coding for antibiotic resistance. Previous studies highlight this same effect, observing how the addition of a strain can mitigate the resistance profile of the edaphic community [26].

Over the past two decades, advances in molecular tools have enabled comprehensive analyses of microbial community composition [66,67,68]. These technological advances have enabled an unprecedented understanding of the taxonomic composition, behavior, response, and ecology of microorganisms [69]. Metagenomics has thus become a fundamental tool for evaluating the impact of different treatments, both chemical and biological, in soil environments [70,71,72,73]. In line with this, amplicon sequencing of the 16S ribosomal RNA gene has enabled the characterization of the microbial community structure and taxonomic diversity [74,75].

Moreover, establishing a causal link between the observed plant responses and the application of specific PGPB strains requires the integration of metagenomic approaches, which are now routinely employed in this field. In line with other authors, it is observed that the type of irrigation can exert a very relevant influence on the taxonomic diversity of the soil community [76]. This influence may vary depending on the stability of the microbial ecosystem, the physicochemical conditions, or alterations to which it is exposed, as well as the nature and granulometry of the soil matrix that contains them. For this reason, although the addition of the strains C1 (*Bacillus pretiosus*) and C2 (*Pseudomonas agronomica*) modifies the taxonomic representation of the genera *Bacillus* and *Pseudomonas*, the taxonomic diversity is not significantly altered, since there is no displacement of other taxa originally represented in the community. The increased relative abundance of the inoculated taxa following biofertilizer application indicates that the introduced PGPB strains persist in the soil throughout the treatment period. The adaptation of the strains to the ecosystem does not alter the taxonomic composition of the soil microbial communities that host them without producing displacements of the resident microbiota. It is foreseeable that once the addition of biofertilizers ceases, microbial communities will recover the representation of these taxa according to their initial state [77,78].

In addition, the observed differences in plant nutritional profiles appear to be associated with the persistence of PGPB strains in their rhizospheres. Various metagenomic studies investigating the effects of PGPB on plants have aimed to elucidate why these bacteria induce changes despite being applied at low microbial concentrations. Consistent with this, it has been shown that the significant differences observed in treatments with added strains arise because both the rhizobiome and the endophytic microbial community play equally important roles in the complex plant–microbe interactions [75].

## 5. Conclusions

The present study has evidenced that the PGPBs C1 (*Bacillus pretiosus*) and C2 (*Pseudomonas agronomica*) have demonstrated their efficacy both in increasing the biomass and in the length of *Quercus pyrenaica plants.* Moreover, the use of organic WWTP fertilizer and EDAR_ST increases biomass production compared to traditional irrigation. Additionally, the incorporation of PGPB (*Pseudomonas agronomica* and *Bacillus pretiosus*) into organic matrices, such as WWTPs, significantly improves the nutritional quality of plants.

Taxonomic analysis confirms the persistence of strains delivered by the biofertilizers in the rhizosphere throughout the treatments, identifying them as key drivers of the observed changes in the host plants. Likewise, the participation of PGPBs in these communities reduces the prevalence of their MIC compared to the most widely used antibiotics, which decreases the potential transmissibility of resistance.

Therefore, having observed the increase in plant biomass, the enrichment of the soil components, and the nutritional quality of the plants subjected to fertigation with WWTP biofertilizers to which *Pseudomonas agronomica* and *Bacillus pretiosus* were added, this study proposes their use in the recovery of soils that favors a more efficient colonization of the land. Further studies with other plant models and in field studies will reveal the potential biotechnological use of the strains and the biofertilizer in forest restoration projects.

## Figures and Tables

**Figure 1 biology-14-00902-f001:**
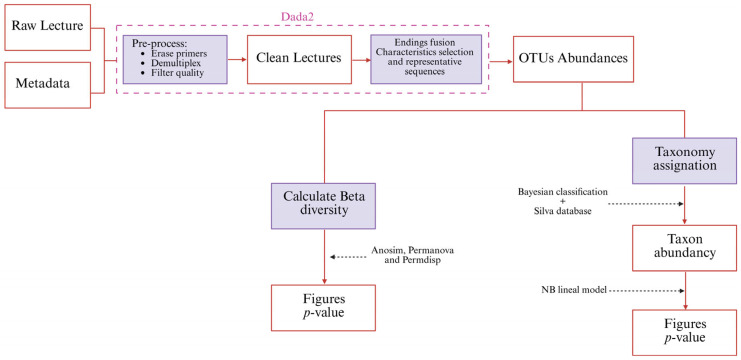
Sequence processing and analysis sequence.

**Figure 2 biology-14-00902-f002:**
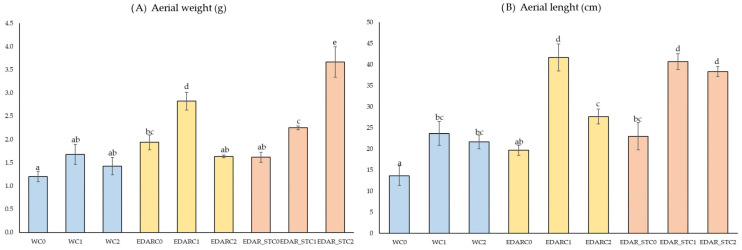
Biometric analysis of (**A**) dry weight of the stem (g) and (**B**) length of the stem (cm). Representation of mean values (n = 3). Bars with identical letters indicate that the average values are not significantly different (*p*-value < 0.05). C1: *Bacillus pretiosus*. C2: *Pseudomonas agronomica*. W: watering with water. EDAR_ST: irrigation with sterilized waste.

**Figure 3 biology-14-00902-f003:**
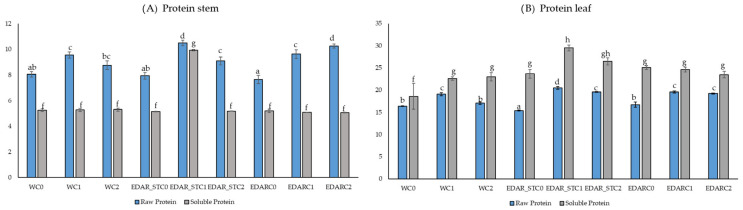
Nutritional factors related to the quality of stem and leaf protein. Mean data for n = 3. Bars with identical letters indicate that the average values are not significantly different (*p*-value < 0.05). %DM (percentage of dry matter). C0: no inoculum; C1: *Bacillus pretiosus*. C2: *Pseudomonas agronomica*. (**A**) Protein parameters in stem; (**B**) protein parameters in leaf.

**Figure 4 biology-14-00902-f004:**
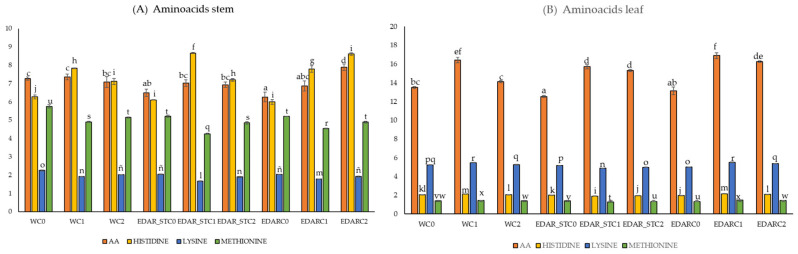
Nutritional factors related to protein quality. Values represent means ± standard deviations (n = 3). Bars sharing the same letter are not significantly different (*p* < 0.05), based on Tukey’s HSD post hoc test. C1: *Bacillus pretiosus*; C2: *Pseudomonas agronomica*; W: watering with water. (**A**) Protein parameters in stem. Letter codes indicate comparisons among means for total amino acids (AA) [a–d], histidine [f–j], lysine [l–o], and methionine [q–u].; (**B**) protein parameters in leaf. Letter codes indicate comparisons among means for total amino acids (AA) [a–f], histidine [i–m], lysine [n–r], and methionine [u–x].

**Figure 5 biology-14-00902-f005:**
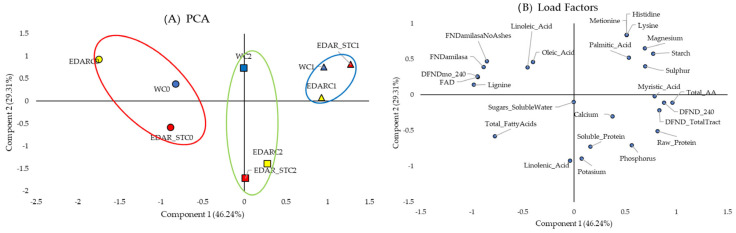
Principal component analysis of the leaf’s nutritional factors. (**A**) PCA, which represents in the (2D) plan the distribution and variation trends of chemical and biological irrigation treatments, according to the two components (nutritional variables of the leaf), which best explain the model. C0: non-inoculated control. C1: *Bacillus pretiosus*. C2: *Pseudomonas agronomica*. W: water irrigation. Blue forms are for water (W) irrigated; yellow forms are for WWTP (EDAR) irrigation, and yellow forms are for sterilized WWTP irrigation (EDAR_ST). Surrounded in red, a grouping of the treatments irrigated with the controls (C0-dots). Surrounded in blue, a grouping of the treatments irrigated with *Bacillus pretisosus* (C1-triangles). Surrounded in green, a grouping of the treatments irrigated with *Pseudomonas agronomica* (C2-squares). (**B**) Load factors.

**Figure 6 biology-14-00902-f006:**
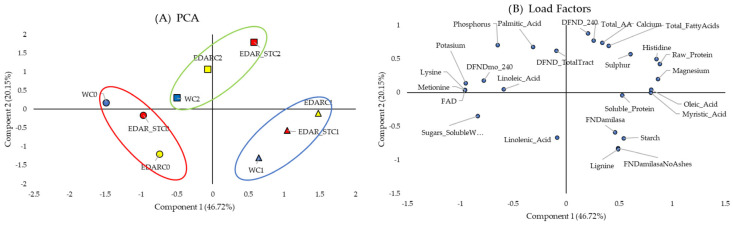
Principal component analysis of the stems’ nutritional factors. (**A**) PCA that represents in the (2D) plan the distribution and variation trends of chemical and biological irrigation treatments, according to the two components (stem nutritional variables), which best explain the model. C0: control without inoculum. C1 (triangles): *Bacillus pretiosus*. C2 (squares): *Pseudomonas agronomica*. W (dots): watering with water. Blue forms are for water (W) irrigated; yellow forms are for WWTP (EDAR) irrigation, and yellow forms are for sterilized WWTP irrigation (EDAR_ST). Surrounded in red, a grouping of the treatments irrigated with the controls (C0). Surrounded in blue, a grouping of the treatments irrigated with *Bacillus pretisosus* (C1). Surrounded in green, a grouping of the treatments irrigated with *Pseudomonas agronomica* (C2). (**B**) Load factors.

**Figure 7 biology-14-00902-f007:**
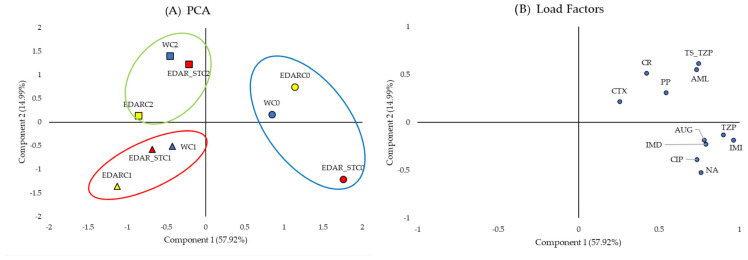
Analysis of the principal components of the CMIs of the edaphic communities. (**A**) PCA, which represents in the (2D) plan the distribution and variation trends of chemical and biological irrigation treatments, according to the two components (variables: antibiotics), which best explain the model. C0: control without inoculum. C1 (triangles): *Bacillus pretiosus*. C2 (squares): *Pseudomonas agronomica*. W (dots): watering with water. Blue forms are for water (W) irrigated; yellow forms are for WWTP (EDAR) irrigation, and yellow forms are for sterilized WWTP irrigation (EDAR_ST). Surrounded in red, a grouping of the treatments irrigated with the controls (C0). Surrounded in blue, a grouping of the treatments irrigated with *Bacillus pretisosus* (C1). Surrounded in green, a grouping of the treatments irrigated with *Pseudomonas agronomica* (C2). (**B**) Load factors.

**Figure 8 biology-14-00902-f008:**
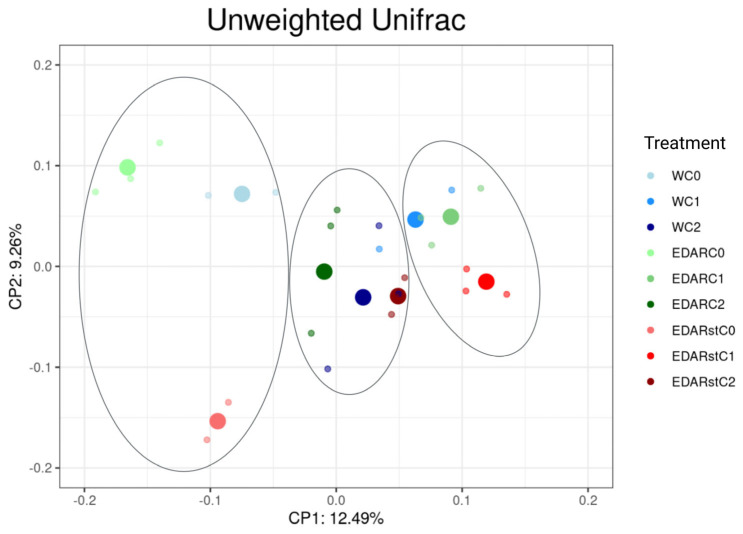
ACP that represents in the plane (2D) the distribution and trends of variation of the taxonomic diversity (*Unweighted Unifrac*) of the samples, depending on the chemical and biological treatment, which best explain the model. C0: non-inoculated control. C1: *Bacillus pretiosus*. C2: *Pseudomonas agronomica*. W: water irrigation. Surrounded in grey, grouping tendencies by biological treatment.

**Figure 9 biology-14-00902-f009:**
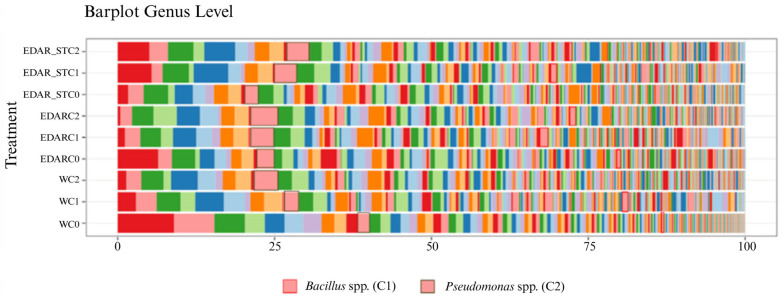
Relative abundances of taxonomic composition at the genus level of rhizospheric samples of *Q. pyrenaica* under the different chemical and biological treatments. C0: non-inoculated control. C1: *Bacillus pretiosus*. C2: *Pseudomonas agronomica*. W: water irrigation.

**Figure 10 biology-14-00902-f010:**
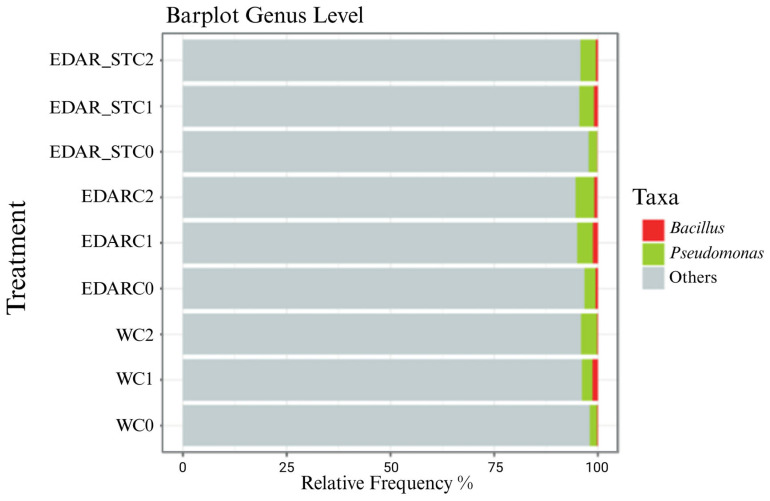
Extract of the relative abundance of biological treatments applied in irrigation (biofertilizer). In red C1 (*Bacillus* sp.), in green C2 (*Pseudomonas* sp.), C0 (without biological treatment).

**Table 1 biology-14-00902-t001:** PGP characteristics of the tested strains.

Code	Identification (WGS)	Origin Insulation	IAA (µg·mL^−1^)	ACCd (p/a)	Siderophores (p/a)
SAICEU11	*Bacillus pretiosus*	*Medicago sativa*	5.61 ± 0.03	−	+
SAICEU22	*Pseudomonas agronomica*	Bulk soil	5.85 ± 0.09	+	+

AIA: production of 3-indoleacetic acid; DCCA: production of 1-aminocyclopropane-1-carboxyl deaminase; P/A: presence (+)/absence (−). WGS: whole genome sequencing.

**Table 2 biology-14-00902-t002:** Physicochemical composition of the WWTP waste. Analysis carried out by LabAqua. Tests covered by ENAC accreditation nº109/LE 28.

	Parameters	Methods	Results	Units
Physico-chemical characteristics	Conductivity at 20 °C	A-F-PE-0015 Electrometry	1454	µS/cm
	Conductivity at 25 °C	A-F-PE-0015 Electrometry	1612	µS/cm
	Biochemical Oxygen Demand (BOD5)	A-F-PE-0002 Manometric Method	3200	mg O_2_/L
	Chemical Oxygen Demand	A-F-PE-0003 Digestion–Colorimetry	6720	mg O_2_/L
	Decanted chemical oxygen demand	A-F-PE-0003 Digestion–Colorimetry	4220	mg/L
	Nitrites	A-F-PE-0010 Digestion	<0.05	mg/L
	Kjeldhal Nitrogen	A-F-PE-0007 Kjeldhal	296.7	mg/L
	pH	A-F-PE-0010 Electrometry	6.5	U. pH.
	Suspended solids	A-F-PE-0006 Gravimetry	3228	mg/L
	Toxicity	PIT-F/0012 Bioluminescence assay with *Vibrio fisheri*	14	U.T.

Majority cations	Potassium	A-D-PE-0025-ICP-OES	60.2	mg/L

Anions	Nitrates	A-BV-PE-0001 HPLC–Conductivity	<2.5	mg/L
	Orthophosphates	Ca-R-PE-0011 Spectrometry	74.32	mg PO_4_/L
	Sulphates	A-BV-PE-0001 HPLC–Conductivity	89.0	mg/L
	Sulphites	A-F-PE-0040 Volumetry	4.5	mg/L

**Table 3 biology-14-00902-t003:** Functional diversity (Shannon’s index, H’) of the soil microbial community at 144 h of incubation.

	WC0	WC1	WC2	EDAR_STC0	EDAR_STC1	EDAR_STC2	EDARC0	EDARC1	EDARC2
H’ 144 h	4.86 ± 0.04	4.13 ± 0.06	4.88 ± 0.04	4.59 ± 0.02	4.68 ± 0.02	4.47 ± 0.01	4.90 ± 0.03	4.72 ± 0.13	4.92 ± 0.03

**Table 4 biology-14-00902-t004:** Quality sample of the taxonomic community.

Metric	Sample
Number of samples	27
Number of Features	10,910
Total frequency	633,769

**Table 5 biology-14-00902-t005:** Frequency per taxonomic sample.

Metric	Frequency
Minimum frequency	1772
First quartile	19,134
Median Frequency	22,096
Third quartile	27,321
Maximum Frequency	54,530
Medium Frequency	23,472.93

## Data Availability

The metagenomic data are in the NCBI database under the accession number PRJNA1154929.

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
