# Peer review of "Use of Bacillus pretiosus and Pseudomonas agronomica for the Synthesis of a Valorized Water Waste Treatment Plant Waste as a Biofertilizer Intended for Quercus pyrenaica L. Fertigation"

_biology, 2025, doi:10.3390/biology14070902_

Round 1
Reviewer 1 Report
Comments and Suggestions for Authors
The article is devoted to the problem of loss of forest areas and the most effective ways of their restoration. The possibility of using organic chemical matrices recovered from WWTP waste organic fertilizers together with promoting bacteria (PGPB) such as Bacillus pretiosus and Pseudomonas agronomica was considered. Harvesting, total biomass measurement study and nutritional analysis were conducted. The change in the rhizosphere bacterial community was also studied. Statistical analysis of biometrics and plant nutrition performed.
Manuscript can be published without correction. There is one request.
- In the chapter objects and methods you describe variants of experiments using organic waste from a WWTP and the same sterilized waste (EDAR_ST). In your charts you use abbreviations EDAR and EDAR_ST. The abbreviation WWTP is not explained anywhere, and EDAR is missing.
Author Response
- In the chapter objects and methods you describe variants of experiments using organic waste from a WWTP and the same sterilized waste (EDAR_ST). In your charts you use abbreviations EDAR and EDAR_ST. The abbreviation WWTP is not explained anywhere, and EDAR is missing.
Revised throughout the manuscript and added the information
Reviewer 2 Report
Comments and Suggestions for Authors
Reviewer comments:
- It is preferable to modify the title, as the scientific and common names should not be written together. It is preferable to write the scientific name only, whether in the title or the entire manuscript. It is suggested that the title be as follows: “Use of Bacillus pretiosus and Pseudomonas agronomica for the synthesis of a valorised WWTP biofertilizer intended for Quercus pyrenaica fertigation”
- In addition; “WWTP” No acronyms allowed in the title.
- L11-15: “The oak (Quercus spp) is ……and humid summers. The state of …. depletion due to regrowth. It should be shortened to one short opening sentence.
- L15: “PGPB” No acronyms when they appear first. Plz define all acronyms when they appear first. If they appear once, there is no need to use acronyms.
- The abstract lacks the methodological part and what is related to its novelty and importance of the current manuscript. The abstract needs to further highlight innovation.
- The abstract is too wordy and not giving clear message to reader, plz revise the abstract section and enriched it with numerical values of study findings (Plz, numerical data or percentages should be added to the most important results obtained from the current study, because it shows the reader the importance of the research and its results).
- The abstract contains a poor conclusion and here authors should add a brief conclusion.
- The keywords are very poor, and they must be attractive and should be different from the title. It is not necessary to number (put numbers in front of each keyword) according to the journal instructions. Plz modify accordingly.
- In the entire introduction section, you should focus only on the topic of the manuscript and what serves this topic, without mentioning many sentences that are not related to the topic or do not serve the idea of ​​the manuscript. The entire introduction section should be revised accordingly.
- The authors need to state the hypothesis in the introduction.
- The authors should emphasize the novelty of the work in the introduction.
- L141: “10 cm x 8 cm” modify to “10 x 8 cm”
- L144 and in the entire manuscript: “Quercus pyrenaica (oak)” write the scientific name only.
- L147: (Community of Madrid, 2024)??
- L156: “3.1. Preparation of bacterial suspensions” if possible, plz add relevant reference.
- L260: “Figure 1” Plz write the figure numbers in parentheses, throughout the manuscript
- Although the results section is well written, it is criticized for being long and repetitive in some parts. There are also transactions and terms that were previously abbreviated and written in full in the results section with their abbreviations. This makes the scientific writing poor. For example, the authors abbreviated the treatments (Bacillus pretiosus) and C2 (Pseudomonas agronomica) to C1and C2, and in every place, they are written in full with their abbreviations. This is incorrect. Rather, the correct thing to do is to write the abbreviations only, such as C1and C2.
- L460: plz start the discussion with the importance and novelty of the current manuscript, then a brief description of each result of the current study, then discuss it, then support the discussion with relevant recent studies, and so on throughout the entire discussion section regarding the different parameters studied.
- Plz discuss the possible mechanism behind your results. Plz Provide more in-depth mechanistic discussion.
- Conclusion: It is interesting, but should be very crisp and not in a detailed form. Plz mention how the future study can complete your work. What is the lack of knowledge?
- L588: “we” Personal pronouns should not be used throughout the manuscript; avoid writing them out.
- L596: “Solanum Lycopersicum L” the scientific name of the entire manuscript should be italic.
- L711: “Arabidopsis Thaliana”, L743: “Bacillus Spp”, same, plz pay attention to this comment in the entire references section.
- If possible, references should be updated with recent studies (past 1-3 years).
The English could be improved to more clearly express the research.
Author Response
- It is preferable to modify the title, as the scientific and common names should not be written together. It is preferable to write the scientific name only, whether in the title or the entire manuscript. It is suggested that the title be as follows: “Use of Bacillus pretiosus and Pseudomonas agronomica for the synthesis of a valorised WWTP biofertilizer intended for Quercus pyrenaica fertigation”
Done
- In addition; “WWTP” No acronyms allowed in the title.
Changed to: “Water Waste Treatment Plant”
- L11-15: “The oak (Quercus spp) is ……and humid summers. The state of …. depletion due to regrowth. It should be shortened to one short opening sentence.
Changed to (lines 11-13): “The oak (Quercus spp) is a tree that grows in acidic and mature soils of the Mediterranean region, especially in sub-Mediterranean or sub-Atlantic environments. The state of conservation of many oak groves is seriously threatened and deteriorated”
- L15: “PGPB” No acronyms when they appear first. Plz define all acronyms when they appear first. If they appear once, there is no need to use acronyms.
Changed to: “Plant Growth Promoting Bacterium”
- The abstract lacks the methodological part and what is related to its novelty and importance of the current manuscript. The abstract needs to further highlight innovation.
Due to the length of the abstract indicated by the journal, we decided to write about three aspects:
- Conceptual framework within which the work was carried out.
- A short brief of material and methods in general terms.
- Short brief of results and general conclusion.
- The abstract is too wordy and not giving clear message to reader, plz revise the abstract section and enriched it with numerical values of study findings (Plz, numerical data or percentages should be added to the most important results obtained from the current study, because it shows the reader the importance of the research and its results).
After looking in detail which data add to the abstract, we think that we can’t talk about the numerical data because if we do that, we are going to exceed the length allowed by the journal for the abstract. We test 9 conditions for the matrixes and the inoculum in each parameter measured, which means that we have a big amount of results that is impossible to brief in the abstract.
- The abstract contains a poor conclusion and here authors should add a brief conclusion.
Changed to (lines 48-48): For all these reasons, the use of the biofertilizer result of the combination of WWTP waste, Bacillus pretiosus and Pseudomonas agronomica is postulated as an environmentally friendly strategy that can contribute to the recovery of potential oak forest areas.
- The keywords are very poor, and they must be attractive and should be different from the title. It is not necessary to number (put numbers in front of each keyword) according to the journal instructions. Plz modify accordingly.
Deleted numbers. Keywords changed to: Biofertilizers; Valorization; WWTP; PGPB; Quercus pyrenaica; Antibiotic resistance; Cenoantibiogram; Reforestation and Degraded soil.
- In the entire introduction section, you should focus only on the topic of the manuscript and what serves this topic, without mentioning many sentences that are not related to the topic or do not serve the idea of ​​the manuscript. The entire introduction section should be revised accordingly.
In the introduction section we talk about the problem of deforestation of Q. pyrenaica in the environment where the study was carried out. As well, we present the potential use of PGPB and biofertilizers in the reforestation process. In the same way we put on the table the importance of reuse of waste products to formulate these biofertilizers and the reintroduction of these products in a circular economy system.
I other hand we concern about the importance of the antibiotic spread in the environment, its potential risk to human health and how there are tools to control it. A concern when we are talking about put exogenous bacteria and waste product in the environment.
At last, we talk about how omics can help us to understand the behaviour of the communities where the biofertilizers are used, link the phenotypic plant response with the action of the inoculum and to follow the fingerprint of our bacteria in the edaphic environment.
All topics that we treat in our manuscript and whose we present results and conclusions.
- The authors need to state the hypothesis in the introduction.
Added in lines 131-134: Therefore, the present study aims to evaluate prospectively, and prior to its field trial, the efficacy of the biofertilizer resulting from the combination of B. pretiosus and P. agronomica strains on Q. pyrenaica seedlings. In the same way, a study of the impact of biofertilizer and strains on rhizospheric communities of inoculated plants was carried out.
- The authors should emphasize the novelty of the work in the introduction.
The present work shows the results of the biotechnological application of two bacterial strains (recently described) and the valorisation/reuse of a waste from a WWTP for the development of a biofertilizer. As well, this work show haw this two strains, which comes from agronomic uses, have a potential use in reforestation and forestry uses. All this information it’s already on the introduction and in the hypothesis.
- L141: “10 cm x 8 cm” modify to “10 x 8 cm”
Done
- L144 and in the entire manuscript: “Quercus pyrenaica (oak)” write the scientific name only.
Done.
- L147: (Community of Madrid, 2024)??
Typo. Deleted.
- L156: “2.3.1. Preparation of bacterial suspensions” if possible, plz add relevant reference.
Uricult trio it’s a dipslide commonly used to indirect evaluation of microbialcrobial growth in liquid samples. These dipslides interpretation it’s on the instructions of each commercial kit.
- L260: “Figure 1” Plz write the figure numbers in parentheses, throughout the manuscript
Done throughout the manuscript
- Although the results section is well written, it is criticized for being long and repetitive in some parts. There are also transactions and terms that were previously abbreviated and written in full in the results section with their abbreviations. This makes the scientific writing poor. For example, the authors abbreviated the treatments (Bacillus pretiosus) and C2 (Pseudomonas agronomica) to C1and C2, and in every place, they are written in full with their abbreviations. This is incorrect. Rather, the correct thing to do is to write the abbreviations only, such as C1and C2.
Changed throughout the results section, leaving the C1 and C2 bacterial names on the figure foots.
- L460: plz start the discussion with the importance and novelty of the current manuscript, then a brief description of each result of the current study, then discuss it, then support the discussion with relevant recent studies, and so on throughout the entire discussion section regarding the different parameters studied.
In the discussion section we follow the same structure that can be seen in all the manuscript. We introduce the conceptual framework of the study and discuss the results and the methods used in the same order as it’s presented on the result section. This is a common form to present the discussion of the results as well as it’s an easy form to follow the ideas while avoids to repeat information along the manuscript.
- Plz discuss the possible mechanism behind your results. Plz Provide more in-depth mechanistic discussion.
We have revised the entire discussion section and consider that the evidence provided is sufficiently described and justified with bibliography. We are at your disposal for any further clarification that may be required
- Conclusion: It is interesting, but should be very crisp and not in a detailed form. Plz mention how the future study can complete your work.
Conclusions revised and added the potential of future studies (looking at the results obtained we are in the early stages of field studies).
What is the lack of knowledge?
The potential use of this tow strains (recently described) in forestry and reforestation uses.
- L588: “we” Personal pronouns should not be used throughout the manuscript; avoid writing them out.
Check throughout the manuscript.
- L596: “Solanum Lycopersicum L” the scientific name of the entire manuscript should be italic.
The italics on the bibliography section are missed caused by the bibliographic manager. Using the bibliographic format according to the Authors Guide and on of the bibliographic managers recommended by the journal (Zotero in our case) the italics appears in common writing and it change in every reboot of the bibliography.
- L711: “Arabidopsis Thaliana”, L743: “Bacillus Spp”, same, plz pay attention to this comment in the entire references section.
Same as above.
- If possible, references should be updated with recent studies (past 1-3 years).
Some of the old references are related with the technique described on it. All the references revised.
